behaviour/ecology/evolution

camouflage, cryptic, sociality, shoaling, Gobiidae

**Author for correspondence:**
Stella A. Encel
e-mail: stella.encel@sydney.edu.au

# Social context affects camouflage in a cryptic fish species

## Stella A. Encel and Ashley J. W. Ward

School of Life and Environmental Sciences, University of Sydney, Sydney, Australia

(iD) SAE, 0000-0001-8677-5920; AJWW, 0000-0003-0842-533X

Crypsis, or the ability to avoid detection and/or recognition, is an important and widespread anti-predator strategy across the animal kingdom. Many animals are able to camouflage themselves by adapting their body colour to the local environment. In particular, rapid changes in body colour are often critical to the survival of cryptic prey which rely on evading detection by predators. This is especially pertinent for animals subject to spatio-temporal variability in their environment, as they must adapt to acute changes in their visual surroundings. However, which features of the local environment are most relevant is not well understood. In particular, little is known about how social context interacts with other environmental stimuli to influence crypsis. Here, we use a common cryptic prey animal, the goby (*Pseudogobius* species 2) to examine how the presence and body colour of conspecifics influence the rate and extent to which gobies change colour. We find that solitary gobies change colour to match their background faster and to a greater extent than gobies in pairs. Further, we find that this relationship holds irrespective of the colour of nearby conspecifics. This study demonstrates the importance of social context in mediating colour change in cryptic animals.

## 1. Introduction

Throughout the animal kingdom, prey animals are under intense selective pressure to develop, and effectively employ, various means of avoiding predation. Consequently, a wide variety of strategies exist to mitigate the risk of predation. Defences often involve some form of crypsis; perhaps the most widely used of all anti-predator strategies. Crypsis, as defined by Ruxton, is a suite of phenotypic and behavioural strategies which augment the user's ability to evade detection by predators [1]. As such, crypsis encompasses an array of approaches including camouflage, nocturnality and refuging behaviour.

Camouflage refers to strategies in which an animal is able to avoid detection by appearing visually indistinct from the

background it is viewed against. Camouflage itself comprises various interrelated means of achieving this outcome [1]. One such means is that of background matching, in which an animal relies on body coloration and/or patterning that closely resembles their background [2,3]. Such strategies are generally most effective when specialized to a specific background or set of backgrounds against which the animal is most often viewed [4,5]. However, prey animals living in environments of higher spatio-temporal variability, or those with more mobile lifestyles, are often subject to a greater degree of environmental heterogeneity and may thus need a more flexible approach to camouflage in order to maintain crypsis [5,6]. Further, a more generalist or plastic approach to camouflage may allow for the ability to exploit a wider variety of habitats, potentially conferring additional fitness-related opportunities [7]. In the light of this, species may employ dynamic approaches such as the ability to rapidly change their body colour using specialized nerve cells known as chromatophores [3,8,9]. Chromatophores are most often located in the dermis and epidermis and are able to effect acute, reversible, neurally mediated changes in external body coloration through the motile activity of chromatosomes [10,11]. Chromatosomes are pigment-containing organelles within chromatophores which may be quickly concentrated or dispersed through the perikaryon of the chromatophore in order to alter the absorption spectra of the cell, cumulatively producing changes in body pattern and coloration [8,10]. Perhaps the most impressive examples of this are seen in colour-changing cephalopods; however, this ability is also widespread among the world's 20 000 species of fishes.

Animals across all taxa are known to use socially acquired information to assist in decision-making related to foraging [12–14], reproduction [15–17] and threat-sensitive behaviours [18,19]. When making decisions, animals must assess the relative benefits and risks of the various options available to them. As such, the ability to collect and integrate relevant information from multiple sources may be of great benefit to animals attempting to avoid predation [20,21]. Many animals, including cryptic prey species, live at densities which allow them to observe one or more conspecifics for much of the time. This provides opportunities to gather social information, which may be used to influence various decision-making processes including those which shape patterns of cryptic behaviour. For animals capable of colour change, such information may influence this process, as shown in some species including shoaling fishes [22]. Social assortment according to body colour has been demonstrated in a number of species of shoaling fish, with individuals preferring to associate with those of similar body colour [23–25]. However, the role of social context in mediating the process of colour change and background matching in camouflaging animals is a neglected area of research. In the only study of which we are aware that has examined this relationship, Chinarina [22] showed that solo cod (*Gadus morhua*) adapt their body coloration to that of their background, but in the presence of others will conform to the colour of shoalmates even if the shoalmates' body colour contrasts with their background. However, that study, published in 1959, was conducted without the benefit of modern colour analysis techniques. Further, no studies have comparatively examined how the presence and body colour of conspecifics moderate the rate and extent of colour change.

The potential ways in which rates of colour change may be affected by social context are not well explored. One possibility is that animals exposed to a sudden change in the background may change colour to match their background faster in the presence of conspecifics than on their own. This could conceivably occur as the result of a socially mediated sensory positive feedback mechanism [25,26] or alternatively as a means to reduce the likelihood of being subject to the oddity effect, in which being visually conspicuous relative to other members of the group makes an individual more vulnerable to predation [27,28]. By contrast, it is also plausible that such an animal might exhibit a slower rate of change in the presence of conspecifics, presuming that the risks of being poorly background-matched are diluted by the presence of other individuals [29,30], and that the process of colour change incurs physiological costs [25,31,32]. Despite the ways in which social information and colour change may interact to influence animals' overall anti-predator strategy, neither of the relevant hypotheses mentioned here have been tested as far as we are aware.

The goby, *Pseudogobius* 'species 2' (hereafter 'goby'), is a small, benthic amphidromous fish widespread throughout coastal waters including estuaries and lagoons from southern Queensland to the south coast of Victoria. Although not known to be a shoaling species, the goby is non-territorial and typically occurs at high densities throughout its range. For instance, at Narrabeen Lagoon, where the animals for this experiment were sourced, this species occurs at densities of approximately 20–50 fish per square metre on mud, sand or gravel flats in shallow areas of the lagoon (A.J.W.W., personal observation). As such, the fish must adapt their coloration to a wide range of substrates within their habitat, from pale sand to very dark mud as well as heterogeneous backgrounds such as variegated gravel. It also forms an important part of the diet of a number of larger fish, including juvenile

barramundi, Australian bass and gudgeons, as well as various wading birds. Camouflage forms a fundamental element of their predator aversion strategy, achieving crypsis by changing their body colour according to their background. This makes them an ideal study species to address the hypothesis that gobies will adjust the rate and extent to which they change colour according to (a) the presence or absence of conspecifics and (b) the body colour of nearby conspecifics.

# 2. Methods

## 2.1. Study species and holding conditions

The goby *Pseudogobius* species 2 (part of a species complex undergoing taxonomic resolution) was used as the model species. Members of this genus are widely distributed throughout temperate, sub-tropical and tropical waters of the Indo-Pacific, occurring in shallow freshwater, marine and estuarine habitats [33]. The range of species 2 extends from southern Queensland to Victoria. It grows to a maximum body size of approximately 35 mm. Due to the absence of any clear sexual dimorphism, adults of both sexes were used for experimentation. Like many fish species, this goby is capable of changing its body colour in response to variation in the physical environment. As is common among gobiids, this species lacks a swim bladder and is adapted to a benthic lifestyle spent in close association with the substrate. This species is non-burrowing and does not typically rely on refugia, instead relying on camouflage. It demonstrates a characteristic saltatory pattern of locomotion, with relatively long periods of stillness punctuated by short hops forward. Fish were collected from a field site at Middle Creek, Narrabeen (33.718491° S, 151.270281° E) between August and October 2020 using large hand-held nets. Fish were captured in shallow waters approximately 5–20 cm in depth. Permits for capture were obtained from the NSW Department of Primary Industries (P16/0135). Following capture, the fish were transported in oxygenated water to holding facilities at the University of Sydney. There, they were housed in tanks containing substrate composed of natural sand and variegated gravel, intended to represent a typical physical environment commonly encountered in the wild. They were fed daily with commercially available fish food (Nutrafin Tropical Flakes, Hagen Products, Germany) and their health was monitored prior to testing.

## 2.2. Experimental design

### 2.2.1. Treatment structure

Fish were exposed to one of two acclimation conditions (white or black background) and one of two test conditions (again, white or black background), for a total of four unique colour treatments. Additionally, fish were tested under one of two social treatments: a solitary treatment, in which the fish was tested alone, and a group treatment, in which the fish were tested in pairs. In the group treatments, fish were exposed to all four colour treatments as well as being either matched or mismatched to their partner. Consequently, there were four treatments where fish were tested alone, and six further treatments where fish were tested in groups, resulting in a total of 10 treatments altogether (table 1). Each of the 10 treatments was replicated six times to a total sample size of $N = 60$.

### 2.2.2. Experimental apparatus

Test arenas were constructed from acrylic plastic and comprised a central square chamber and two removable side chambers with sliding door mechanisms allowing entry to the central chamber. The central arena measured $20 \times 20 \times 20$ (L × W × H), while the side chambers measured $6 \times 6 \times 15$ (L × W × H). There were two arenas, one black and one white, and four side chambers, two black and two white. The removable side chambers allowed pairing of any combination of colours (e.g. two black, two white, or one of each) with each colour of test arena. When side chambers were not matched to the colour of the central test arena, black or white duct tape was used to cover the part of the side chamber that faced into the central arena. The whole arena was placed inside a large plastic tub, and a frame was built around the entire apparatus. This frame was covered in white Corflute® in order to minimize external disturbance. Within the screens, four LED light batons (cool white, 3300 K) were placed, two at each end of the arena along the short axes of the tub. Lights were positioned so as to prevent direct illumination of the test arena and instead provide diffuse light.

**Table 1.** Treatment structure.

| treatment number | social treatment | acclimation colour treatment | | test colour treatment |
|---|---|---|---|---|
| 1 | solitary | black | | black |
| 2 | solitary | black | | white |
| 3 | solitary | white | | black |
| 4 | solitary | white | | white |
| 5 | paired | black | | black |
| 6 | paired | black | black | white |
| 7 | paired | black | white | black |
| 8 | paired | black | white | white |
| 9 | paired | white | white | black |
| 10 | paired | white | white | white |

## 2.3. Experimental protocol

For each replicate, fish were haphazardly selected from the holding aquaria and transferred to the test apparatus in beakers. The fish were initially added to the side chambers (maximum one fish per side chamber) and allowed to acclimate for 15 min. This timing was based on pilot studies conducted prior to the experiment described here, which showed that most colour change occurs within 2–3 min, so this 15 min acclimation was conservative. Once this period had elapsed, fish were released (simultaneously, in the case of pairs) into the central test arena by manually raising the sliding door of each side chamber. After the fish had exited the side chamber, the door was closed behind it. Fish were then allowed to move freely around the arena for a period of 15 min. Each replicate was filmed using a Canon G1X camera positioned approximately 50 cm above the central arena. The camera was set to take a sequence of still images, rather than to video, since the former allowed a higher resolution of 4352 × 3264 pixels per image. The frequency of images was approximately one image per half-second in white tested treatments and approximately one image per 1.25 s in black tested treatments. This was due to the longer exposure period required in the darker (black) arena. No flash was used. After testing, fish were removed and transferred to a separate holding tank. No fish were re-used.

## 2.4. Analysis

From the sequence of images, we selected the first image following the entry of the fish to the arena. Subsequently, we selected images at 30 s intervals. Where this was not possible due to blurred images as a result of test subject movement, we selected the closest possible alternative in the series of images. Where a suitable image could not be selected within ±10 s of the scheduled sampling point, the data point was disregarded and the next image was taken from the subsequent sampling point. This, however, was a rare occurrence, accounting for five data points out of 600 across all trials.

### 2.4.1. Colour analysis

Adobe Photoshop was used to measure the RGB values of each fish and the background against which it was tested over time. The RGB colour model is an additive three-dimensional model for the mathematical and digital representation of colour, based on a trichromatic visual system in which the visual spectrum is perceived as an additive combination of red, blue and green light. In this model, colours can be represented within a three-dimensional plane by a set of coordinates with values ranging from 0 to 255. A standard 24-bit RGB colour display devotes 8 bits per pixel to each spectral channel (red, green, blue) in order to construct colour images. A binary-encoded system ($2^8 = 256$) thus permits values from 0 to 255 for each spectral channel, with 255 representing full saturation. As such, in the RGB colour model, $(0, 0, 0)$ denotes pure black and $(255, 255, 255)$ denotes pure white. This system allows the expression of millions of unique shades as Cartesian points within the three-

dimensional RGB colour plane ($256^3 = 16\,777\,216$). On this basis, the difference between any two colours can be quantified by calculating the Euclidean three-dimensional distance between their RGB coordinates. In order to quantify the difference between a fish and its background, the magnetic lasso tool was used to trace the outline of the fish. After this, the fish could be cut and pasted onto a transparent background to allow the colour of the fish to be measured independent of its background. The eyedropper tool was then used to measure the average RGB values across the entire body of the fish. Similarly, the eyedropper tool was used to measure the RGB values of the background within one body length of the fish.

Together, these measures were used to calculate the three-dimensional Euclidean distance between the colour of the fish and the background colour over time.

### 2.4.2. Statistical analysis

All analyses were conducted using R. Data were visually inspected using QQ plots and histograms. Outliers were identified using Cook's distance. In total, three trials of the total 60 were removed due to anomalous coloration in the test subjects. Since there were differences in the ability of gobies to conform to black versus white backgrounds (see electronic supplementary material), we conducted the analyses for these separately. To test whether the presence (or absence) of conspecifics, and the background colour to which those conspecifics had been acclimated, affected the rate and extent of colour change of gobies, we used a linear mixed-effects model from the lme4 package for R. The response variable was the difference between gobies and their background. Fixed effects were treatments and time (as a continuous variable). For white acclimated gobies against a black background, we used treatments 3, 7 and 9 (table 1). For black acclimated gobies tested against a white background, we used treatments 2, 6 and 8. In each case, therefore, we compared gobies tested on their own, gobies tested in a pair (where the gobies used in the analysis were acclimated to a different colour from their partner) and gobies tested in a pair (where both were acclimated to the same colour). Using model selection methods (model.sel from the MuMIn package), we determined that the colour-changing response of fish as a function of time followed a quadratic function, rather than a linear function. Finally, we specified trial as the random effect to account for multiple observations being taken within each trial.

## 3. Results

When tested against a background which did not match their acclimation colour, gobies changed colour to converge with the test background, typically within 2–4 min. When tested against a background which did match their acclimation colour, there was no change in the gobies' colour. These relationships were independent of social treatment (see electronic supplementary material).

White acclimated gobies tested against a black background changed colour as a function of the interaction between treatment and time ($\chi^2 = 6.498$, d.f. $= 2$, $p = 0.039$; figure 1$a$). The rate and extent of colour change shown by solitary gobies was greater than that of gobies tested in pairs. Treatment as a fixed effect in the model was a significant predictor of colour change ($\chi^2 = 11.009$, d.f. $= 2$, $p = 0.004$), as was time ($\chi^2 = 69.247$, d.f. $= 1$, $p < 0.001$).

Similarly, black acclimated gobies tested against a white background also changed colour as a function of the interaction between treatment and time ($\chi^2 = 9.467$, d.f. $= 2$, $p = 0.009$; figure 1$b$). Again, the rate and extent of colour change shown by solitary gobies was greater than that of gobies tested in pairs. Treatment as a fixed effect in the model was not a significant predictor of colour change ($\chi^2 = 3.932$, d.f. $= 2$, $p = 0.14$), while time was significant ($\chi^2 = 44.169$, d.f. $= 2$, $p < 0.001$).

## 4. Discussion

Colour change in cryptic gobies is influenced by social context, in particular the presence or absence of nearby conspecifics. Specifically, gobies tested in the absence of conspecifics changed colour more rapidly and to a greater extent than those tested in the presence of conspecifics. Interestingly, the colour of nearby conspecifics had no significant effect on the colour change of focal gobies. As expected, gobies were better able to adapt their body colour to a darker than a lighter background.

Solitary gobies showed a faster and more pronounced change in colour than those tested alongside nearby conspecifics. To our knowledge, this is the first demonstration that social context influences the

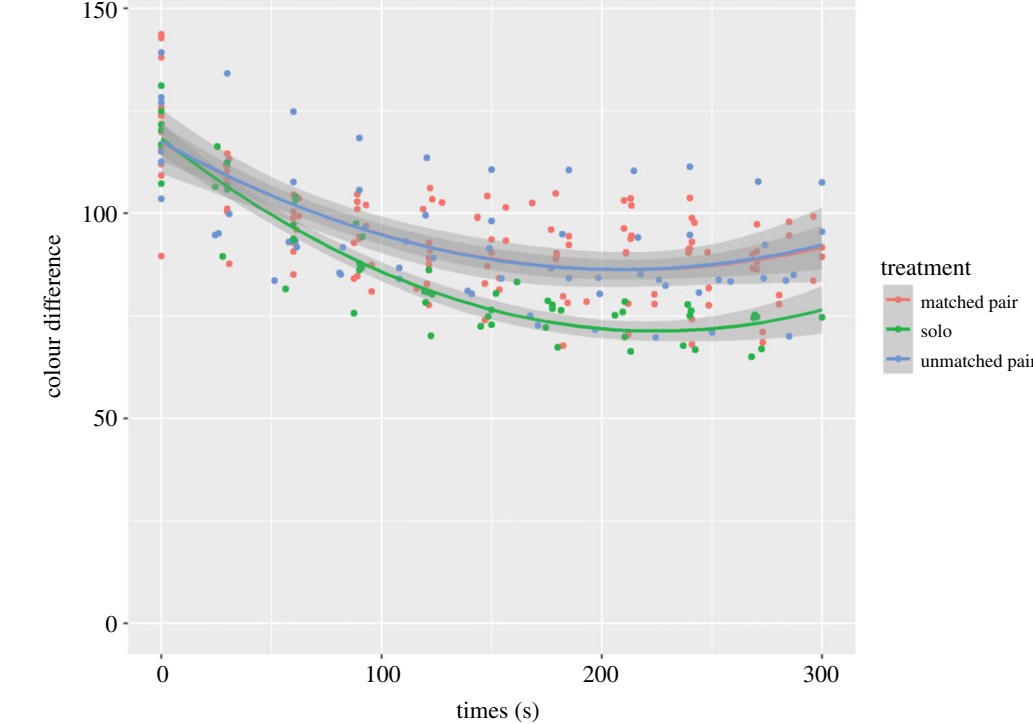

**Figure 1.** Change in colour of gobies as a function of time (in seconds) in (*a*) white acclimated gobies tested against a black background and (*b*) black acclimated gobies tested against a white background. Green line indicates single individuals, red line indicates colour-matched pairs, blue line indicates colour-mismatched pairs. Colour-matched pairs refer to treatments where both partners were acclimated to the same background. Colour-mismatched pairs refers to treatments where one fish was acclimated to a white background and one to black. In colour-mismatched pairs, only the individual tested against the alternative background to that which they were acclimated is included in the graph. Shaded areas represent 95% CI.

rate of colour change in a cryptic animal. Although a previous study has shown that the body colour of nearby conspecifics influences the process of colour change, this study highlights the importance of social context in mediating the rate and extent of colour change in a cryptic animal. There are a number of

potential explanations for this. Although these gobies do not exhibit shoaling behaviour or social attraction with that of nearby conspecifics [34], they are non-territorial and occur at densities which often place them in proximity to conspecifics. As such, they may benefit from risk dilution, which in turn may influence their anti-predator behaviour, including crypsis. The increased rate and greater extent of colour change in solo versus grouped fish may be driven by differences in stress levels experienced by gobies in the two grouping treatments. Individuals in groups are known to benefit from risk dilution, in which predation risk is negatively correlated with the number of other individuals present [35,36]. This is part of the broader phenomenon of social buffering, whereby the stress experienced by individuals is ameliorated by the presence of others [37–39]. This is widespread among social animals across all taxa and although it is rarely, if ever, tested in non-grouping animals, it seems reasonable to imagine that a similar mechanism would apply presuming that the species in question is tolerant of conspecifics, as is the case with the gobies used here. Physiological manifestations of acute stress in vertebrates are known to be mediated by the sympathetic nervous system in general and the adrenergic system in particular, relying predominantly on the neurochemical activity of norepinephrine [40]. Norepinephrine has also been shown to occupy an essential role in facilitating the cytological mechanisms which underpin the process of rapid colour change [31,41,42]. Consequently, it is possible that the rate and extent of colour change are moderated by stress. Were this the case, the reduced rate and extent of change seen in groups may be explained by decreased adrenergic activity associated with lower levels of acute stress, due to the alleviation of stress afforded by the effects of social buffering. Additionally, this trend may be underpinned by the existence of a trade-off between the perceived risk of predation and the physiological costs incurred by the process of colour change. Animals capable of colour change have demonstrated the ability to modulate background matching according to ambient predation risk [43]. It is also well established that the cellular mechanisms underlying the process of rapid colour change involve the motile activity of chromatosomes, a neurally mediated process that is necessarily accompanied by metabolic costs [8,31,41]. In the light of this, the notion that the dilution of risk and/or social buffering effects afforded by the presence of conspecifics may allow individuals to allay the metabolic costs associated with colour change by moderating the rate and extent of this process seems plausible.

In contrast with a previous study [22], we found that colour change was not affected by the body colour of near neighbours. Specifically, gobies paired with a partner of matching colour and gobies that were paired with a partner of a contrasting colour each changed colour to match their background at a similar rate. This may be to do with the social tendency of the subjects, as the study which found that the body colour of nearby conspecifics influenced colour change used Atlantic cod (*G. morhua*), a shoaling species. By contrast, the gobies used here do not exhibit grouping tendencies, and as such may have a lower propensity to rely on cues arising from conspecifics across a variety of contexts [34], including colour change. Another possible explanation for this is the tendency of gobies, unlike the species used in the previous study, to associate closely with their substrate and to move relatively infrequently, compared to species occupying higher positions in the water column. The implication of this is that gobies are in closer proximity to their background, and this becomes the dominant factor in determining colour change. The present study examines rapid physiological colour change, on the basis that adapting to short-term shifts in background coloration is of critical importance in maintaining crypsis and avoiding predation. This response is thought to be governed by two separate mechanisms [44], the first of which is the so-called primary light response, wherein chromatophores independently effect a change in colour in response to direct stimulation by light; however, this response is most readily observed in larval fish rather than adults. The secondary light response, which is slower both in ontogenetic development and acute manifestation, is a neurohormonal response mediated largely by visual input and tends to dominate the primary light response. Thus, when adapting body colour to match the environment, an animal should theoretically integrate the colour of the various objects which fall within their visual field, including other animals, to adopt an appearance which is the best approximation of their immediate surroundings. This being the case, the presence of an individual which contrasts the background may have a minimal effect on the overall visual field, especially when the prevailing background is (as in this case) largely homogeneous. Future work should examine the role of heterogeneity in both the abiotic background and the colour of nearby fish, particularly including higher densities of near neighbours, in shaping an individual's ability to adaptively match to its background.

Individuals tested against light backgrounds showed a consistent reduction in the degree to which they were able to conform to their background, compared to those tested against dark backgrounds. While individuals tested against light backgrounds achieved a minimum colour distance of

approximately 75 units from their background, dark-tested individuals were able to conform to their background within roughly 25 units of colour distance. This may be due to the fact that the amount of light reflected by pale substrates is inherently greater than that by darker substrates, potentially making any object viewed against a light substrate more visible, regardless of its colour. The reduced ability to conform to light coloured backgrounds observed here has also been shown in rock gobies (*Gobius panganellus*). Furthermore, this species has been shown to make decisions regarding microhabitat selection based on this differential ability, preferring dark substrates [45]. The tendency of rock gobies to select parts of their habitat that enhance crypsis is part of the broader phenomenon of risk sensitivity in prey species. Animals are well known to adapt their behaviour in accordance with the perceived risk of predation [46,47]. Aquatic animals in particular may assess local predation risk through the presence of chemical cues known as alarm substances arising from predators and injured conspecifics. For example, fish have been shown to alter social behaviours such as shoaling and group size upon detection of alarm cues [21,33,47]. Variable responsiveness to alarm cues depending on the presence of other individuals has also been demonstrated [48]. This suggests that fish may modulate threat-sensitive behaviours according to both social context and ambient predation risk. As such, it may be beneficial to examine how the rate and extent of colour change vary according to both social context and the perception of ambient predation risk, perhaps indicated by the presence of alarm cues. This could offer insight into how the effects of social buffering and risk perception potentially act together to regulate threat-sensitive behaviours, including those associated with crypsis such as colour change.

Ethics. This work was carried out with the approval of the University of Sydney Animal Ethics Committee (reference no. 2019/1616).

Data accessibility. Data and associated R code have been uploaded to Dryad: https://doi.org/10.5061/dryad.0zpc866xw.

Authors' contributions. S.A.E. designed and performed experiments, conducted analysis and wrote the paper. A.J.W.W. designed experiments, conducted analysis and wrote the paper.

Competing interests. We declare we have no competing interests.

Funding. This work was supported by a grant from the Australian Research Council (grant no. DP190100660).

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
