## [Peer Review File · Royal Society Open Science]

Review History

RSOS-211125.R0 (Original submission)

Review form: Reviewer 1 (Daniel Osorio)

Is the manuscript scientifically sound in its present form?

Yes

Are the interpretations and conclusions justified by the results?

Yes

Is the language acceptable?

Yes

Do you have any ethical concerns with this paper?

No

Have you any concerns about statistical analyses in this paper?

No

Recommendation?

Accept with minor revision (please list in comments)

Comments to the Author(s)

This is an interesting study of the effects of social context on camouflaging behaviour of a goby. The main finding is that the presence of another fish reduces the rate and degree of colour change when the fish is moved from a light to a dark background of vice-versa. This effect is independent of whether the social partner is the same or different in colour. The ms is nicely written, and clearly situates the findings in relevant literature. For a single experiment whose interpretation is unclear the ms is rather long and also repetitious. I recommend a shorter and more focussed Introduction.

Review form: Reviewer 2**Is the manuscript scientifically sound in its present form?**

No

Are the interpretations and conclusions justified by the results?

Yes

Is the language acceptable?

Yes

Do you have any ethical concerns with this paper?

Yes

Have you any concerns about statistical analyses in this paper?

No

Recommendation?

Major revision is needed (please make suggestions in comments)

Comments to the Author(s)

Lines 59-61 (page 4). You seem to have crypsis and camouflage backwards. Actually crypsis is a subset of camouflage. Crypsis avoids or minimizes detection. But camouflage is any way to prevent a predator from recognizing it as an object of interest. Camouflage thus includes crypsis, but it also includes looking like bird droppings, rocks, twigs, leaves or other inedible objects. So please fix this sentence.

Lines 87-107 (pp. 5-6) don't add anything and can be removed. Not really relevant to this paper.

Page 7. It shouldn't take 7 pages to get to the main point of the paper. You can be a lot more concise! You can do it in on or perhaps 2 pages, not 7.

lines 167-168--unknown species. Given the fluid taxonomy you should be very specific about where you collected the fish for testing because that will eventually help to identify which species you studied here. Not enough to say "QLD-VIC" as there could easily be several species over the entire eastern coastline of AU.

Line 180--sorry I see this now. It should go immediately after you say that the species is unidentified.

Line 182--you should give the permit numbers.

Line 194. Why white vs black? Perhaps white sand in your study area, but black seems unrealistic. Why not brown or some other naturally-occurring colour? The black background may generate irrelevant and misleading color changes. If white is brighter than the highest background there will be the same problem. You need to justify these backgrounds. Otherwise a good design. However I could not tell if each of the 4 treatments were replicated. Please explain. Lines 200 (six) and line 201 (ten) contradict each other and the description of the 4 treatments. Please explain the treatments more clearly! One of the reasons it is confusing is that you say 6 when I expect 8 (2 social 2 colors, pre and post acclimation). This is very confusing and needs serious rewriting. At least you finally say that the treatments were replicated on line 201.

Lines 226-227, "haphazardly". Does this mean that fish were used more than once and you cannot tell which fish were used 2 or more times? Do you have pseudoreplication and if so did you allow for that with a GLMM?

Line 235. Is 15 minutes really enough for the fish to settle down? Do you have field data after disturbance to get an idea how long they normally take to settle down after a disturbance? I take it that they don't require shelters like other substrate resting gobies? If they do require shelters and there are none, then the behaviour will be abnormal and conclusions dubious. You need to explain what you did relative to what the gobies do naturally.

Line 256. You are using a human-based color model--this may be entirely irrelevant to goby vision. You need to say something about this. If you are just looking at lightening and darkening it is probably ok, and you did have just a light and a dark background with no color. So this is probably ok for this particular experiment (although gray would be better than black). Given that you are only looking at lightening and darkening there is reason for all that discussion about color. You basically just have a luminance experiment. That is ok, but don't waste time/space on irrelevancies.

Line 320. Please give the times to acclimate on white or black (see my comments above).

Lines 331-332. I think that at least one paper by Devi Stuart-Fox and co-workers also found this in a color-changing lizard. So not exactly a first, but a first with a very careful measure of time and the explicit difference between solitary and social. Please check to retain scientific accuracy.

Lines 354-356. I was very surprised to see this comment about stress, as my first reaction to your results was the tradeoff between social visual signalling and being cryptic. That exactly predicts your results. Of course there could be general stress induced by the lack of shelters, which means that two fish together and neither has a shelter would lead to very strong stress, owing to predation risk in the wild. So you need to discuss both possible causes of your interesting results.

Decision letter (RSOS-211125.R0)

Dear Miss Encel

The Editors assigned to your paper RSOS-211125 "Social Context Affects Camouflage in a Cryptic Fish Species" have now received comments from reviewers and would like you to revise the paper in accordance with the reviewer comments and any comments from the Editors. Please note this decision does not guarantee eventual acceptance.

Please submit your revised manuscript and required files (see below) no later than 21 days from today's (ie 16-Aug-2021) date. Note: the ScholarOne system will 'lock' if submission of the revision is attempted 21 or more days after the deadline. If you do not think you will be able to meet this deadline please contact the editorial office immediately.

on behalf of Professor Cinzia Chiandetti (Associate Editor) and Kevin Padian (Subject Editor)
openscience@royalsociety.org

Associate Editor Comments to Author (Professor Cinzia Chiandetti):

Both reviewers agree that the draft is unnecessarily long and make suggestions as to which parts should be shortened. In addition, one reviewer raises several methodological concerns that I hope you can address in a revised version of the manuscript. The point about shelters is particularly important given the potential stress experienced by the animals.

Reviewer comments to Author:

Reviewer: 1

Comments to the Author(s)

This is an interesting study of the effects of social context on camouflage behaviour of a goby. The main finding is that the presence of another fish reduces the rate and degree of colour change when the fish is moved from a light to a dark background of vice-versa. This effect is independent of whether the social partner is the same or different in colour. The ms is nicely written, and clearly situates the findings in relevant literature. For a single experiment whose

interpretation is unclear the ms is rather long and also repetitious. I recommend a shorter and more focussed Introduction.

Reviewer: 2

Comments to the Author(s)

Lines 59-61 (page 4). You seem to have crypsis and camouflage backwards. Actually crypsis is a subset of camouflage. Crypsis avoids or minimizes detection. But camouflage is any way to prevent a predator from recognizing it as an object of interest. Camouflage thus includes crypsis, but it also includes looking like bird droppings, rocks, twigs, leaves or other inedible objects. So please fix this sentence.

Lines 87-107 (pp. 5-6) don't add anything and can be removed. Not really relevant to this paper.

Page 7. It shouldn't take 7 pages to get to the main point of the paper. You can be a lot more concise! You can do it in on or perhaps 2 pages, not 7.

lines 167-168--unknown species. Given the fluid taxonomy you should be very specific about where you collected the fish for testing because that will eventually help to identify which species you studied here. Not enough to say "QLD-VIC" as there could easily be several species over the entire eastern coastline of AU.

Line 180--sorry I see this now. It should go immediately after you say that the species is unidentified.

Line 182--you should give the permit numbers.

Line 194. Why white vs black? Perhaps white sand in your study area, but black seems unrealistic. Why not brown or some other naturally-occurring colour? The black background may generate irrelevant and misleading color changes. If white is brighter than the highest background there will be the same problem. You need to justify these backgrounds. Otherwise a good design. However I could not tell if each of the 4 treatments were replicated. Please explain. lines 200 (six) and line 201 (ten) contradict each other and the description of the 4 treatments. Please explain the treatments more clearly! One of the reasons it is confusing is that you say 6 when I expect 8 (2 social 2 colors, pre and post acclimation). This is very confusing and needs serious rewriting. At least you finally say that the treatments were replicated on line 201.

Lines 226-227, "haphazardly". Does this mean that fish were used more than once and you cannot tell which fish were used 2 or more times? Do you have pseudoreplication and if so did you allow for that with a GLMM?

Line 235. Is 15 minutes really enough for the fish to settle down? Do you have field data after disturbance to get an idea how long they normally take to settle down after a disturbance? I take it that they don't require shelters like other substrate resting gobies? If they do require shelters and there are none, then the behaviour will be abnormal and conclusions dubious. You need to explain what you did relative to what the gobies do naturally.

Line 256. You are using a human-based color model--this may be entirely irrelevant to goby vision. You need to say something about this. If you are just looking at lightening and darkening it is probably ok, and you did have just a light and a dark background with no color. So this is probably ok for this particular experiment (although gray would be better than black). Given that you are only looking at lightening and darkening there is reason for all that discussion about

color. You basically just have a luminance experiment. That is ok, but don't waste time/space on irrelevancies.

Line 320. Please give the times to acclimate on white or black (see my comments above).

Lines 331-332. I think that at least one paper by Devi Stuart-Fox and co-workers also found this in a color-changing lizard. So not exactly a first, but a first with a very careful measure of time and the explicit difference between solitary and social. Please check to retain scientific accuracy.

Lines 354-356. I was very surprised to see this comment about stress, as my first reaction to your results was the tradeoff between social visual signalling and being cryptic. That exactly predicts your results. Of course there could be general stress induced by the lack of shelters, which means that two fish together and neither has a shelter would lead to very strong stress, owing to predation risk in the wild. So you need to discuss both possible causes of your interesting results.

===PREPARING YOUR MANUSCRIPT===

===PREPARING YOUR REVISION IN SCHOLARONE===

Author's Response to Decision Letter for (RSOS-211125.R0)

See Appendix A.

Decision letter (RSOS-211125.R1)

Dear Miss Encel,

It is a pleasure to accept your manuscript entitled "Social Context Affects Camouflage in a Cryptic Fish Species" in its current form for publication in Royal Society Open Science.

on behalf of Professor Cinzia Chiandetti (Associate Editor) and Kevin Padian (Subject Editor)
openscience@royalsociety.org

Associate Editor Comments to Author (Professor Cinzia Chiandetti):
Associate Editor

Comments to the Author:

The Authors have included all suggestions made by the Reviewers in the revised version of the manuscript by clarifying previous concerns. This increases the understanding and readability of their relevant findings.

Appendix A

Associate Editor Comments to Author (Professor Cinzia Chiandetti):

Both reviewers agree that the draft is unnecessarily long and make suggestions as to which parts should be shortened. In addition, one reviewer raises several methodological concerns that I hope you can address in a revised version of the manuscript. The point about shelters is particularly important given the potential stress experienced by the animals.

Reviewer comments to Author:

Reviewer: 1

Comments to the Author(s)

This is an interesting study of the effects of social context on camouflaging behaviour of a goby. The main finding is that the presence of another fish reduces the rate and degree of colour change when the fish is moved from a light to a dark background of vice-versa. This effect is independent of whether the social partner is the same or different in colour. The ms is nicely written, and clearly situates the findings in relevant literature. For a single experiment whose interpretation is unclear the ms is rather long and also repetitious. I recommend a shorter and more focussed Introduction.

WE THANK THE REFEREE FOR THEIR CONSTRUCTIVE AND USEFUL COMMENTS. WE AGREE WITH THE COMMENTS ABOUT LENGTH AND HAVE SHORTENED THE MANUSCRIPT ACCORDINGLY. SPECIFICALLY, WE HAVE SHORTENED THE INTRODUCTION.

Reviewer: 2

Comments to the Author(s)

Lines 59-61 (page 4). You seem to have crypsis and camouflage backwards. Actually crypsis is a subset of camouflage. Crypsis avoids or minimizes detection. But camouflage is any way to prevent a predator from recognizing it as an object of interest. Camouflage thus includes crypsis, but it also includes looking like bird droppings, rocks, twigs, leaves or other inedible objects. So please fix this sentence.

WHILE WE ARE AWARE THERE IS INCONSISTENCY IN THE USE OF THESE TERMS, IN THE INTRODUCTION WE STATE OUR USE OF THE TERMS CRYPSIS AND CAMOUFLAGE IN LINE WITH THE DEFINITIONS GIVEN IN THE DEFINITIVE TEXT ON THIS SUBJECT (AVOIDING ATTACK, G.D RUXTON). HAVING CLEARLY STATED OUR OPERATING DEFINITIONS WE DO NOT FEEL THAT THIS CAUSES UNDUE CONFUSION.

Lines 87-107 (pp. 5-6) don't add anything and can be removed. Not really relevant to this paper.

REMOVED AS SUGGESTED

Page 7. It shouldn't take 7 pages to get to the main point of the paper. You can be a lot more concise! You can do it in on or perhaps 2 pages, not 7.

MANUSCRIPT HAS BEEN SHORTENED.

lines 167-168--unknown species. Given the fluid taxonomy you should be very specific about where you collected the fish for testing because that will eventually help to identify which species you studied here. Not enough to say "QLD-VIC" as there could easily be several species over the entire eastern coastline of AU.

Line 180--sorry I see this now. It should go immediately after you say that the species is unidentified.

AS IDENTIFIED BY THE REVIEWER WE DO PROVIDE THE SPECIFIC DETAILS OF CAPTURE SITE. IN THIS PARAGRAPH WE MOVE FROM GENERAL INFORMATION ABOUT THE SPECIES TO MORE SPECIFIC INFORMATION ABOUT THOSE USED IN THIS STUDY. AS SUCH, WE FEEL THAT THE TEXT IS SUFFICIENTLY CLEAR IN ITS CURRENT FORM, HOWEVER IF THE EDITOR DISAGREES WE WILL CHANGE IT.

Line 182--you should give the permit numbers.

ADDED

Line 194. Why white vs black? Perhaps white sand in your study area, but black seems unrealistic. Why not brown or some other naturally-occurring colour? The black background may generate irrelevant and misleading color changes. If white is brighter than the highest background there will be the same problem. You need to justify these backgrounds. Otherwise a good design. However I could not tell if each of the 4 treatments were replicated. Please explain.

FIRST, WE CHOSE WHITE VERSUS BLACK AS THESE PROVIDE THE EXTREMES OF THE LUMINANCE SPECTRUM. IN THE WILD, THE GOBIES MAY BE FOUND IN SHALLOW WATER ON VERY LIGHT COLOURED SAND, VERY DARK SILT, OR ANY INTERMEDIATE COLOUR. BASED ON OUR OBSERVATIONS, THE COLOUR DIFFERENCE BETWEEN THE LIGHTEST COLOUR NATURAL SUBSTRATES AND WHITE, AND BETWEEN THE DARKEST COLOURED NATURAL SUBSTRATES AND BLACK ARE NOT EXTENSIVE.

SECOND, YES, ALL OF OUR TREATMENTS WERE REPLICATED. WE STATE THIS IN THE 'TREATMENT STRUCTURE' SUBSECTION OF THE METHODS.

lines 200 (six) and line 201 (ten) contradict each other and the description of the 4 treatments. Please explain the treatments more clearly! One of the reasons it is confusing is that you say 6 when I expect 8 (2 social 2 colors, pre and post acclimation). This is very confusing and needs serious rewriting. At least you finally say that the treatments were replicated on line 201.

WE PROVIDE A TABLE THAT SETS OUT ALL OF THE TEN DIFFERENT TREATMENTS CLEARLY. WE HAVE REWORDED THIS PARAGRAPH.

Lines 226-227, "haphazardly". Does this mean that fish were used more than once and you cannot tell which fish were used 2 or more times? Do you have pseudoreplication and if so did you allow for that with a GLMM?

FISH WERE TAKEN HAPHAZARDLY FROM THEIR HOLDING AQUARIA, AS DESCRIBED. WE ALSO STATE THAT NO FISH WERE REUSED.

Line 235. Is 15 minutes really enough for the fish to settle down? Do you have field data after disturbance to get an idea how long they normally take to settle down after a disturbance? I take it that they don't require shelters like other substrate resting gobies? If they do require shelters and there are none, then the behaviour will be abnormal and conclusions dubious. You need to explain what you did relative to what the gobies do naturally.

BASED ON PILOT STUDIES, 15 MINUTES WAS SUFFICIENTLY LONG FOR THE FISH TO ADAPT TO THE COLOUR OF THEIR ARENAS. THEY DO NOT USE SHELTERS EITHER IN THE WILD OR IN CAPTIVITY, RELYING INSTEAD ON CAMOUFLAGE. THE GOBIES TEND TO BE FOUND IN THE WILD ON OPEN FLAT SUBSTRATES WHERE THE AVAILABILITY OF REFUGIA IS EXTREMELY LIMITED OR NON EXISTENT.

Line 256. You are using a human-based color model--this may be entirely irrelevant to goby vision. You need to say something about this. If you are just looking at lightening and darkening it is probably ok, and you did have just a light and a dark background with no color. So this is probably ok for this particular experiment (although gray would be better than black). Given that you are only looking at lightening and darkening there is reason for all that discussion about color. You basically just have a luminance experiment. That is ok, but don't waste time/space on irrelevancies.

WE ARE PRIMARILY LOOKING AT LIGHTENING AND DARKENING. IT'S ALSO IMPORTANT TO NOTE THAT THE RGB VALUES OF THE BLACK BACKGROUND DID NOT APPROACH (0,0,0) AND SIMILARLY THE WHITE DID NOT APPROACH (255,255,255), WHICH MEANS THAT IN ESSENCE WE WERE USING SHADES OF GREY, AS YOU SUGGEST. WHICHEVER SCALE IS USED, ARGUMENTS CAN BE MADE AGAINST IT. THE IMPORTANT ASPECT OF THIS IS THAT IT IS OBJECTIVE.

Line 320. Please give the times to acclimate on white or black (see my comments above).

WE PROVIDE THIS IN THE SUPPLEMENTARY MATERIALS AND REPORT ON THIS IN THE MAIN MANUSCRIPT.

Lines 331-332. I think that at least one paper by Devi Stuart-Fox and co-workers also found this in a color-changing lizard. So not exactly a first, but a first with a very careful

measure of time and the explicit difference between solitary and social. Please check to retain scientific accuracy.

THANKS FOR THIS. WE HAVE ADDED THIS CITATION.

Lines 354-356. I was very surprised to see this comment about stress, as my first reaction to your results was the tradeoff between social visual signalling and being cryptic. That exactly predicts your results. Of course there could be general stress induced by the lack of shelters, which means that two fish together and neither has a shelter would lead to very strong stress, owing to predation risk in the wild. So you need to discuss both possible causes of your interesting results.

TO REITERATE, THESE FISH DO NOT USE SHELTERS. MORE BROADLY, WE CONSIDER THAT IT IS IMPORTANT TO GIVE FULL CONSIDERATION TO POTENTIAL EXPLANATIONS FOR OUR FINDINGS.

WE THANK THE REVIEWER FOR THEIR DETAILED COMMENTS.